# Validation of a Patient-Derived Xenograft Model for Cervical Cancer Based on Genomic and Phenotypic Characterization

**DOI:** 10.3390/cancers14122969

**Published:** 2022-06-16

**Authors:** Shunsuke Miyamoto, Tomohito Tanaka, Kensuke Hirosuna, Ruri Nishie, Shoko Ueda, Sousuke Hashida, Shinichi Terada, Hiromi Konishi, Yuhei Kogata, Kohei Taniguchi, Kazumasa Komura, Masahide Ohmichi

**Affiliations:** 1Department of Obstetrics and Gynecology, Educational Foundation of Osaka Medical and Pharmaceutical University, 2-7, Daigaku-machi, Takatsuki 569-8686, Osaka, Japan; shunsuke.miyamoto@ompu.ac.jp (S.M.); ruri.nishie@ompu.ac.jp (R.N.); shouko.ueda@ompu.ac.jp (S.U.); sosuke.hashida@ompu.ac.jp (S.H.); shinichi.terada@ompu.ac.jp (S.T.); hiromi.konishi@ompu.ac.jp (H.K.); yuhei.kogata@ompu.ac.jp (Y.K.); m-ohmichi@ompu.ac.jp (M.O.); 2Translational Research Program, Educational Foundation of Osaka Medical and Pharmaceutical University, 2-7, Daigaku-machi, Takatsuki 569-8686, Osaka, Japan; ompu20171084@s.ompu.ac.jp (K.H.); kohei.taniguchi@ompu.ac.jp (K.T.); kazumasa.komura@ompu.ac.jp (K.K.)

**Keywords:** cervical cancer, DNA, extracellular vesicles, patient-derived xenograft, RNA

## Abstract

**Simple Summary:**

The rate of total tumor engraftment of patient-derived xenografts is 50% in cervical cancer. These cancers retain their histopathological characteristics. The gene mutations and expression patterns associated with carcinogenesis and infiltration and the expression levels of genes in extracellular vesicles released from the tumors are similar between patient-derived xenograft models and primary tumors. Patient-derived xenograft models of cervical cancer could be potentially useful tools for translational research.

**Abstract:**

Patient-derived xenograft (PDX) models are useful tools for preclinical drug evaluation, biomarker identification, and personalized medicine strategies, and can be developed by the heterotopic or orthotopic grafting of surgically resected tumors into immunodeficient mice. We report the PDX models of cervical cancer and demonstrate the similarities among original and different generations of PDX tumors. Fresh tumor tissues collected from 22 patients with primary cervical cancer were engrafted subcutaneously into NOD.CB17-PrkdcSCID/J mice. Histological and immunohistochemical analyses were performed to compare primary and different generations of PDX tumors. DNA and RNA sequencing were performed to verify the similarity between the genetic profiles of primary and PDX tumors. Total RNA in extracellular vesicles (EVs) released from primary and PDX tumors was also quantified to evaluate gene expression. The total tumor engraftment rate was 50%. Histologically, no major differences were observed between the original and PDX tumors. Most of the gene mutations and expression patterns related to carcinogenesis and infiltration were similar between the primary tumor and xenograft. Most genes associated with carcinogenesis and infiltration showed similar expression levels in the primary tumor and xenograft EVs. Therefore, compared with primary tumors, PDX models could be potentially more useful for translational research.

## 1. Introduction

Cervical cancer is one of the most common cancers affecting women worldwide. In 2018, an estimated 570,000 cases of cervical cancer were diagnosed and 311,000 deaths were reported [1]. Cervical cancer has been managed effectively for decades; however, it remains the most common cause of cancer-related death among women [2]. Clinicians and researchers require abundant cancer specimens for precision medicine and the development of new anticancer drugs. Most patients with advanced-stage tumors receive radiotherapy or chemotherapy after biopsy. Surgical therapy may be performed at an early stage for small lesions [3]. Thus, it is difficult to obtain sufficient specimens for examination and research purposes. However, patient-derived xenograft (PDX) models are advantageous for these patients because the tumor tissue can be expanded in mice to generate adequate samples for tumor tissue analysis [4]. Preclinical studies using animal models are essential for the pathological analysis of malignancies and the development of new therapeutic agents. Less than 5% of new cancer treatments in the market have been approved, even though preclinical studies have been successful [5]. The lack of an appropriate human cancer model is one of the reasons for this slow approval rate. Animal models transplanted with human cell lines do not always accurately represent human cancer pathology or definite drug responses. PDX models that preserve the main characteristics of the original tumor are increasingly being used in preclinical and translational research [6]. PDX models, which are new animal models, are established by the heterotopic or orthotopic grafting of fresh surgically resected tumor tissues into immune-deficient mice [7,8]. PDX models reproduce the clinicopathological characteristics of the original tumor and are used as experimental models for drug evaluation, biomarker identification, and precision medicine strategies [6]. Several PDX models have been established successfully, including colon [9], stomach [10], breast [11], pancreas [12], lung [13], liver [14], kidney [15], bladder [16], uterus [17], and ovary [18]. However, a limited number of studies have reported PDX models of cervical cancer. The purpose of this study was to determine whether the characteristics (including pathological findings, gene mutations, and gene expression) identified in the primary tumor were consistent with those of the corresponding PDX tumor identified using PDX models of cervical cancer. To the best of our knowledge, this is the first report on the comparison of PDX models of cervical cancer with primary tumors based on genetic analysis data.

## 2. Materials and Methods

### 2.1. Patients and Tissue Samples

This study enrolled 22 patients between February 2018 and January 2021, who were diagnosed with cervical cancer. The patients underwent laparoscopic or abdominal radical hysterectomy at Osaka Medical and Pharmaceutical University in Japan. During surgery, fresh tumor tissues were collected and divided into three portions. The first portion was immediately placed in Dulbecco’s modified Eagle’s medium nutrient mix F-12 (DMEM/F12, Gibco, Thermo Fisher Scientific, Tokyo, Japan) mixed with Matrigel (Corning, New York, NY, USA) on ice for transplantation, the second portion was fixed in 10% formalin for pathological analysis, and the third portion was placed in RNAlater tissue storage reagent (Thermo Fisher Scientific) for gene analysis.

### 2.2. Animals

All animal experiments were approved by the Osaka Medical and Pharmaceutical University Pharmaceutical Ethics Committee (Assurance Number 21007-A). In this study, 4–8-week-old female NOD.CB17-PrkdcSCID/J mice (Oriental BioService, Kyoto, Japan) were used for the implantation of human cervical cancer tissue. These animals were housed in a specific pathogen-free barrier facility at 24–26 °C with a humidity of 30–50% and free access to sterile water and standard rodent chow. If animal sacrifice was required, trained staff performed cervical dislocation and euthanasia.

### 2.3. Pathological Analysis with Immunohistochemistry

Tissue sections from primary and PDX tumors were prepared using standard procedures and stained with H&E. For immunohistochemical analysis, P16 (ab54210, 1:100 dilution, Abcam, Cambridge, MA, USA) and Ki-67 (M7240, 1:100 dilution, Dako Japan, Tokyo, Japan) antibodies were used to confirm that the PDX tumor was derived from the original tumor. After incubation with primary antibodies, the slides were rinsed with PBS, incubated with a species-specific secondary antibody, and stained using 3,3′-diaminobenzidine substrate solution (MBL, Osaka, Japan). Images were obtained using a microscope (BZ-X700 Series, Keyence, Osaka, Japan). The Ki-67 labeling index was determined by counting 500 tumor cells in the hotspots of nuclear labeling. Ki-67-positive cells were counted in the area with the highest proliferative activity [19].

### 2.4. Establishment of PDX

Isoflurane gas anesthesia was administered to minimize pain and movement during the procedure. Tumor tissue was chopped into fragments (3 mm^3^), mixed with Matrigel (Corning), and then injected into the subcutaneous tissue in the dorsal region of immunodeficient mice using a 22-gauge needle. The mice that underwent transplantation were examined once a week. When the backs of the mice were swollen and engraftment was confirmed, the mice were euthanized (Figure 1). The xenograft tumor tissues were divided into several fragments for analysis. The tissues were used for preparing the next-generation PDX. The remaining tissues were cryopreserved at −80 °C for long-term storage.

### 2.5. Preparation of Extracellular Vesicles (EVs)

Tissue samples were immediately immersed in 4 mL of DMEM/F12 (Thermo Fisher Scientific) containing 10% FBS (Gibco, Grand Island, NY, USA) and stored at 48 °C for 3 h. The tissue-soaked medium was centrifuged at 2000× *g* for 30 min and filtered through a 0.22 μm filter (Merck Millipore, Bedford, MA, USA) to remove cell debris. To recover EVs, ultracentrifugation was performed at 100,000× *g* for 90 min using Optima XE-100 (Beckman Coulter, Brea, CA, USA), SW41 T1 (Beckman Coulter), and Ultra-Clear tubes (Beckman Coulter). EVs were collected in PBS (Figure 2a). EVs were confirmed using Western blotting, nanoparticle tracking analysis (NTA), and electron microscopy.

### 2.6. Western Blot Analyses

Western blotting was performed as described previously [20,21,22]. Briefly, EV samples were lysed with Laemmli SDS-sample buffer with or without 2-mercaptoethanol. Protein samples were separated by SDS-PAGE and transferred to polyvinylidene difluoride membranes. The membranes were blocked with 10% BSA in 1X TBS and incubated overnight with specific primary antibodies against CD63 (1:1000 dilution; Santa Cruz Biotechnology, Dallas, TX, USA) and CD9 (1:1000 dilution; COSMO BIO, Tokyo, Japan) at 4 °C. After washing, the membranes were incubated with a mouse immunoglobulin secondary antibody for 1 h. Bands were visualized using an enhanced chemiluminescence agent (ECL Plus; GE Healthcare Life Sciences, Pittsburgh, PA, USA) (Figure 2b).

### 2.7. NTA

NTA measurements were performed using NanoSight NS 300 (Quantum Design Japan, Tokyo, Japan). The collected EV pellets were resuspended in 1 mL of PBS and diluted at a 1:300 ratio before analysis. The samples were loaded into the instrument and analyzed according to the manufacturer’s instructions using NTA software version 3.4 (Quantum Design Japan) (Figure 2c).

### 2.8. Scanning Electron Microscopy

EVs were incubated with poly-L-lysine-solution-coated beads (φ 3.10 μm; Merck Millipore). After drying, the beads were washed and fixed in 1.25% glutaraldehyde in 0.1 M phosphate buffer (pH 7.4). The beads were washed again with phosphate buffer and fixed with 1% osmium tetroxide for 40 min. After washing, the beads were gradually dehydrated using a series of stepwise ethanol washes. Platinum–palladium was evaporated on the surface of specimens, which were further examined with a scanning electron microscope (S-5000; HITACHI, Tokyo, Japan) (Figure 2d).

### 2.9. DNA and RNA Extraction

According to the manufacturer’s protocol, the MagMAX DNA Multi-Sample Ultra 2.0 and MagMAX mirVana Total RNA Isolation Kits (Thermo Fisher Scientific) were used to extract genomic DNA and total RNA from primary tumors, PDX (F0 and F2) tumors and both types of EVs. DNA and total RNA were quantified using the Qubit dsDNA and RNA high-sensitivity assay kits and a Qubit Fluorometer (Thermo Fisher Scientific).

### 2.10. Amplicon Sequencing

Amplicon sequencing for gene mutation analysis was performed using the Ion AmpliSeq Cancer Hotspot Panel v 2 (Thermo Fisher Scientific). It was designed to examine approximately 2800 COSMIC mutations in 50 oncogenes and tumor suppressor genes. The Ion GeneStudio S5 series was used to sequence the libraries. We then used the Torrent Suite software to analyze the automated cancer hotspot variant. The software is available on the IonGene Studio S5 Series Trent Server. We used the IonReporter software 5.10 to annotate variants and consolidate information from multiple databases. Tumor-specific somatic mutations with mutation allele frequency >10% were detected, and filtered mutation data were used to detect gene mutations.

### 2.11. RNA Sequencing

RNA sequencing was performed for gene expression analysis using an Ion AmpliSeq RNA Cancer Panel (Thermo Fisher Scientific). It was developed as an RNA complement for the Ion AmpliSeq Cancer Hotspot Panel v2. The panel is a single pool of primers that targets 50 oncogenes and tumor suppressor genes, including KRAS, BRAF, and epidermal growth factor receptor (EGFR). Templates were created using the Ion One Touch 2 system. After sequencing the RNA-derived amplicons on the IonTorrent sequencing platform, the number of reads mapped to each gene was counted to determine the expression level of the target gene present in the sample. This strategy enables the comparison of the relative expression levels of target genes among different samples.

### 2.12. Data Analyses

Sequencing data were processed using the Torrent Suite Software v5.12.1 and Ion Reporter 5.0 software (Thermo Fisher Scientific). In addition, the data were processed using a script written in the programming language Python v3.9.5. Filter-based annotation in ANNOVAR (24 October 2019) was used for variant annotation. Paplot v0.5.5 and matolotlib/seaborn were used as drawing tools. Supercomputing resources were provided by the Human Genome Center, Institute of Medical Science, University of Tokyo.

### 2.13. Statistical Analyses

Statistical analyses were performed using JMP Pro version 14.2.0 (SAS Institute Japan, Tokyo, Japan). Continuous variables are expressed as medians (interquartile range). The Mann–Whitney U-test was used to compare continuous variables, and Fisher’s exact test was used to compare frequencies. Pearson’s correlation coefficient (denoted as R) was used to determine the concordance rate between each pair. Statistical significance was set at *p* < 0.05.

## 3. Results

### 3.1. Establishment of the PDX Cervical Cancer Model

An overview of the clinical characteristics of the patients and a comparison of their tumorigenicity are presented in Table 1. The success rate of the transplants was 50% (11/22). The median age of patients was 48.5 years (33–70 years) and 13.6% of the patients (3/22) had Federation of Gynecology and Obstetrics (FIGO) stage III cervical cancer. The engraftment rates were higher in patients with large tumors (≥4 cm), high serum squamous cell carcinoma antigen and carbohydrate antigen 125 levels, and advanced FIGO stages than in patients without them; however, the differences were not statistically significant (*p* = 0.1, 0.1, and 0.8). The median follow-up was 17.5 (21–12) months, and 5 of the 11 patients whose tumors were used to successfully establish a PDX model showed recurrence. However, none of the 11 patients, whose tumors could not be used to establish a PDX model, experienced recurrence. Therefore, we can infer that tumors with clinically poor prognoses have a high PDX engraftment rate.

### 3.2. Histological Evaluation of Patient and PDX Mouse Tumors

Figure 3 shows the pathological findings with the immunohistochemistry of primary tumors and PDXs. Pathologically, four primary tumors with squamous cell carcinoma (PDX 71, 75, 81, and 99) had structural and morphological features similar to those of the PDX. Immunohistochemical analysis showed similar expression profiles of proteins (p16 and Ki-670) in primary tumors and xenografts. Most cancer cells in the primary tumor and PDX showed strong expression of p16. The Ki-67 labeling indices in the primary tumor and PDX were 66% and 48% in PDX 71, 35% and 49% in PDX 75, 61% and 72% in PDX81, and 34% and 55% in PDX 99, respectively.

### 3.3. Genomic Profiling of Primary and PDX Tumors

The overlap of functional mutations, including frameshift deletion, frameshift insertion, non-frameshift substitution, non-frameshift single nucleotide variant, and stop-gain mutations between primary and PDX mouse tumors, was analyzed in 10/11 established PDX tumors (Figure 4a). The most commonly mutated genes were FMS-like tyrosine kinase 3 (FLT3), Erb-B2 Receptor Tyrosine Kinase 4 (ERBB4), and Cyclin Dependent Kinase Inhibitor 2A (CDKN2A). In all cases, the majority of somatic mutations were observed in both primary and PDX tumors. The concordance rates for somatic mutations between primary and F0 mouse tumors were 80.0% in PDX 71, 60.0% in PDX 75, 70.0% in PDX 81, 68.8% in PDX 99, 78.6% in PDX 126, 81.8% in PDX 151, 92.9% in PDX 160, 83.3% in PDX161, 69.2% in PDX 231, and 61.5% in PDX 251. The concordance rates were consistent for all cases. The concordance rates for somatic mutations between primary and F2 mouse tumors were 56.3% in PDX 99 and 57.1% in PDX 126, which were lower than those between primary and F0 tumors. Figure 4b shows the variant allele frequencies (VAFs) of the somatic mutations identified in both primary and PDX tumors. Pearson’s correlation coefficients between primary and F0 tumor VAFs were 0.994 for PDX 71, 0.927 for PDX 75, 1.00 for PDX 81, 0.915 for PDX 151, 0.927 for PDX 160, 0.934 for PDX 161, 0.999 for PDX 231, and 0.997 for PDX 251. In contrast, it was −0.455 for PDX 99, and 0.402 for PDX 126. Strong similarities were observed in the VAFs of somatic mutations between primary and PDX tumors in eight out of ten samples.

### 3.4. RNA Expression Analysis of Primary and PDX Tumors

Figure 5a shows the gene expression of the primary and PDX tumors using heatmap analysis. In the hierarchical clustering analysis, upregulated and downregulated genes are color-coded red and blue, respectively. Sequence samples of the primary and PDX tumors (F0 and F2) are labeled red, blue, and green, respectively. Overall, the primary and PDX tumors showed similar profiles. High- and low-expression groups were clustered. In particular, the CDKN2A, GNAS, and NPM1 genes were expressed at high levels, whereas the HNF1A, ALK, and FLT3 genes were expressed at low levels. Gene expression was analyzed using the AmpliSeq RNA Cancer Panel to determine whether the profiles were preserved in the PDX tumors. In PDX 71, 75, 81,151, 160, 161, 231, and 251, the Pearson’s correlation coefficients for the expression of tumor genes between primary and F0 tumors were 0.898, 0.768, 0.910, 0.669, 0.837, 0.808, 0. 852, and 0.907, respectively (Figure 5b). RNA sequencing revealed that gene expression in PDX (F0) tumors correlated well with that in primary tumors. In PDX 99 and 126, the Pearson’s correlation coefficients for the expression of tumor genes between primary and F0 tumors were 0.804 and 0.739, respectively. In contrast, those between P and F2 were 0.653 and 0.699, respectively, which were lower than those between P and F0 (Figure 5c), suggesting that the correlation of gene expression between primary and PDX tumors decreased with the passage number.

### 3.5. RNA Expression Analysis of EVs in Primary and PDX Tumors

Figure 6a shows the gene expression in the EVs of primary and PDX tumors using heatmap analysis. In the hierarchical clustering analysis, upregulated and downregulated genes were color-coded red and blue, respectively. Sequence samples of primary tumor EVs (P-EV) and PDX tumor EVs (F0-EV and F2-EV) were labeled red, blue, and green, respectively. As expected, EVs from primary and PDX tumors showed similar profiles. High- and low-expression groups were clustered. In particular, the CDKN2A, GNAS, and NPM1 genes were expressed at high levels, whereas the RET, ALK, and FLT3 genes were underexpressed. Gene expression was analyzed using the AmpliSeq RNA Cancer Panel to determine whether the profiles were preserved in the EVs of the PDX tumors. In PDX 99, 126, and 151, the Pearson’s correlation coefficients for the expression of tumor genes between P-EV and F0-EV were 0.766, 0.714, and 0.734, respectively (Figure 6b). RNA sequencing revealed that gene expression in PDX (F0) tumor EVs correlated well with that in primary tumor EVs. In contrast, those in P-EV and F2-EV were 0.740 and 0.676, respectively, which were lower than those in P-EV and F0-EV (Figure 6b). The correlation of gene expression between primary and PDX tumor EVs appeared to decrease with passage.

## 4. Discussion

In this study, we observed strong similarities in tumor characteristics, including histological and genomic characteristics associated with tumorigenicity, between the primary and PDX tumors in cervical cancer. PDX models of cervical cancer (CC-PDX) retained the pathological and genomic characteristics of the primary tumor.

The engraftment rate is important for establishing a PDX model because of limited funding and the low availability of tissue specimens. Several factors determine the success rate of PDX. The success rate ranges from 0–75% in published studies [23,24,25,26,27]. Previously, we reported that 61 CC-PDXs were established from tumor tissues obtained from 98 patients with cervical cancer; the overall enrollment rate was 62.2% [23,24,25,26,27]. A tumor fragment of 1–3 mm^3^ may be suitable for establishing CC-PDX [28]. Hoffman et al. reported the implantation of 3–5 mm tumor fragments into SCID mice; however, no viable tumors were observed [23]. In the current study, no association was observed between engraftment rate and clinical characteristics of the patients; however, all adenocarcinomas failed to grow. In general, tumor fragments obtained from patients with a large tumor size or advanced stage have a higher success rate of transplantation than tumors obtained from patients with less-advanced disease [28]. Most established CC-PDX is squamous cell carcinoma [23,24,25,26,27,28]. Several authors created CC-PDX of adenocarcinoma; however, the successful rate of adenocarcinoma was lower than squamous cell carcinoma [23,24,25,26,27,28]. Hoffmann et al. minced tumor pieces with scissors and aspirated into a syringe. The tumor cell suspension was subcutaneously inoculated into SCID mice. The successful rate was 100% in adenocarcinoma and 40% in squamous cell carcinoma [23]. Chaudary et al. implanted the tumor fragment into cervix of SCID mice. They established 16 CC-PDX. No associations were found between the xenograft take rate and the histology. However, most of the CC-PDX, which they established, was squamous cell carcinoma. Among 16 established CC-PDX, adenocarcinoma was 2 [24]. Oh et al. made a subrenal capsula xenograft of a nude mouse with a 1 mm^3^ tumor fragment. Among four adenocarcinomas, two CC-PDXs were established; the successful rate was 50% [26]. Larmour et al. also created a subrenal capsula xenograft. The authors established two CC-PDXs of adenocarcinoma; the successful rate was 100% [27]. These data suggest that minced tumor cells or subrenal capsula xenograft might be suitable for the CC-PDX of adenocarcinoma.

Nude mice are commonly used to establish CC-PDX. However, reportedly, the engraftment rate is higher in severely immunodeficient mice than in nude mice [23,24,29,30,31]. Nude mice have been identified by the appearance of alopecia, but these mice lack T cells because they do not have a thymus [32]. NOD.CB17-PrkdcSCID/J mice lack T cells, B cells, and NK cells. Therefore, the lack of immune cells is considered one of the factors affecting the engraftment rate. Several types of tumor transplantation for establishing CC-PDX based on the site of injection, namely, subcutaneous, subrenal capsule, and orthotopic transplantations, have been reported. Subcutaneous transplantation is the most common procedure because confirmation of the tumor transplant is easy; however, metastasis to other organs rarely occurs [23,25,30,33]. Subrenal capsule transplantation can be used for tumors or normal tissues that are unlikely to be malignant. Although the procedure is complicated, the growth of tumors in the renal capsule increases the blood supply and is expected to have a high engraftment rate [26,27,29]. Orthotopic transplantation is also common because it enables a more accurate reproduction of the tumor environment [24,25]. In this study, 3 mm^3^ tumor fragments were injected subcutaneously into NOD.CB17-PrkdcSCID/J mice with an engraftment rate of 50%.

Most studies on CC-PDX have reported that the characteristic pathological features are maintained in primary as well as PDX tumors [23,24,25,26,33]. Cervical cancer is usually caused by human papillomavirus (HPV) infection. The HPV E7 protein functionally inactivates retinoblastoma (Rb), resulting in p16 overexpression, which is one of the most important characteristics in cervical cancer. Immunohistochemically, similar patterns for p16 expression were observed in primary and PDX tumors [27]. In this study, the tumor cells showed strong expression of p16 in both primary and PDX tumors. Ki67 expression increases with passage [34]. These results confirm that the PDX model maintains the biological properties of the primary tumor and that cell proliferation may be enhanced during passage.

Previously, most published studies on CC-PDX analyzed pathological characteristics to validate primary and PDX tumors [24,25,27]. A study on CC-PDX analyzed gene expression, and the PDX tumor showed faithful reproduction of the gene expression patterns of the primary tumor; however, the target gene was only HER-2 [26]. Zhu et al. performed DNA and RNA sequencing to compare original to F4 PDX tumors in two high-grade endometrioid carcinomas [35]. Most of the mutations were similar in the primary and PDX tumors. Mutation frequencies exhibited a significant linear correlation. In gene expression pattern by RNA sequencing, the expression of genes exhibited a significant linear correlation. Depreeuw et al. performed whole-exon sequencing in grade 1 and 3 endometrioid carcinoma without MSI or POLE mutations [36], and the majority of results were common between primary and PDX tumors. On average, 90% of the genome had the same copy number in the primary and PDX tumors. Bonazzi et al. performed whole-exome sequencing in endometrial cancers, including four common molecular subtypes [37]. They focused on mismatch repair-deficient (MMRd) and p53 mutant subtypes. MMRd models are expected to accumulate changes during passaging based on defective DNA mismatch repair. The authors observed minimal mutational heterogeneity in non-MMRd models and some heterogeneity in MMRd models. In the p53 mutated subtype, the total number of somatic mutations was consistent between primary tumors and PDXs. Cybula et al. performed genomic analysis focusing on single-nucleotide polymorphisms in high-grade serous carcinomas in ovarian cancers [38]. The authors concluded that ovarian PDX lines largely remain stable throughout propagation. However, some marginal genetic drift occurred during PDX initiation. The authors also observed several genetically unstable PDXs potentially associated with DNA repair deficiency owing to BReast CAncer gene mutations. In this study, genomic analysis of 11 PDX lines demonstrated that the PDX library reproduced the genomic characteristics of the primary tumor with high fidelity (Figure 4 and Figure 5). In addition, we also analyzed the mRNAs contained in EVs extracted from primary and PDX tumors (Figure 6). EVs are nanometer-sized endosomal vesicles secreted from various cell types. By transferring their cargo (miRNAs, mRNAs, DNA, and proteins), EVs can affect intercellular communication. Therefore, an increasing number of researchers have focused on the potential usefulness of EVs. Previously, EV-related research focused primarily on exosomes derived from body fluids and cultured cells [21,39]. However, in recent years, reports of EVs directly derived from tissues have been released [22]. Meanwhile, few reports of EVs extracted from PDXs have been published [40]. To the best of our knowledge, this is the first report on EVs derived from PDX tissues. Hierarchical clustering of gene expression profiles revealed that EVs of primary tumors clustered directly with the EVs derived from PDX models. These results also indicate a high degree of similarity between the EVs of primary cancer cells and the corresponding EVs of PDX tumor cells. In summary, we cannot conclude that the PDX model perfectly represents the genetic profile of the patient; however, this study provides evidence that the PDX model is useful for clinical targeted therapy trials and precision medical research.

In the current study, all PDX models contained genetic mutations of NRAS, whereas the primary tumors did not. Mutations in the Ras family of genes occur in a quarter of all human cancers. Downstream signaling through NRAS is critical to diverse cellular processes involved in tumorigenesis, including cell proliferation, metabolism, and survival [41]. The induction of NRAS mutation may be an important characteristic in CC-PDX. It may occur in a reaction of tumor cells through the influence of mouse tissue. Otherwise, the stroma replaced by mouse-derived tissue may affect the same [42].

DNA-seq in CC-PDX and primary tumors showed several mutations, including those in genes encoding FLT3, ERBB4, CDKN2A, Kinase Insert Domain Receptor (KDR), fibroblast growth factor recep-tor 2 (FGFR2), enhancer of zeste homolog 2 (EZH2), SMAD Family Member 4 (SMAD4), ataxia telangiectasia mu-tated (ATM), SWI/SNF Related, Matrix Associated, Actin Dependent Regulator Of Chromatin, Subfamily B, Member 1 (SMARCB1), Serine/threonine kinase 11 (STK11), and RB1. ERBB4 is a member of the EGFR family, comprising four transmembrane tyrosine kinases [43]. FGFR2 belongs to the FGFR family [44]. KDR, known as vascular endothelial growth factor receptor, is a transmembrane tyro-sine kinase and acts as a receptor during vascular endothelial growth [45]. FLT3 is one of the most extensively studied proteins in hematopoietic malignancies. Binding to FMS-like receptors leads to the homodimerization and autophosphorylation of FLT3, resulting in the transduction of pro-survival and proliferative signals through the RAS/MAPK, JAK/STAT, and PI3K/AKT pathways [46]. These transmembrane tyrosine kinases are widely used in breast [47], lung [48], and hematopoietic malignancies [49]. CDKN2A, known as p16, acts as an inhibitor of CDK 4/6 [50]. The compounds CDK4/6 and cyclin D1 release the transcription factor E2F from Rb, which results in the phosphorylation of Rb and subsequent cell proliferation [51]. SMARCB1 (BAF47/INI1) is a subunit of the BRG1/BRM-associated factor (BAF) complex and participates in tumor suppression via the p16-Rb, Wnt, and sonic hedgehog pathways, among others [52]. In cervical cancer, HPV E7 protein inactivates Rb, causing the overexpression of CDKN2A [53]. Molecular targeting therapy using Rb showed no effect in most patients with cervical cancer. EZH2 mediates histone methylation for epigenetic regulation [54]. SMAD4 is a member of the Smad family of transcription factor proteins and is the central signal transducer of the transforming growth factor-beta (TGF-beta) signaling pathway. This signaling pathway is well known for its role in inducing epithelial–mesenchymal transition [55]. The serine/threonine kinase ATM mediates signal transduction following DNA double-strand breaks. PERP inhibitors are widely used for ATM-expression-deficient cancers [56]. We selected the molecular targets described above. Some of them may have efficacy in cervical cancer. Hence, drug tests in the CC-PDX model may be more useful before human therapy.

This study had several limitations. First, the sample size for genomic profiling was relatively small. Second, DNA and RNA sequencing were not performed considering the precipitated mouse tissue. Third, amplicon sequencing was performed for only cancer-related tumor genes; these data did not include other primary genes. Fourth, although several gene mutations that could be a target for therapy were identified, a drug efficacy test was not performed. In consideration of these points, further examinations, including drug efficacy tests, are needed.

## 5. Conclusions

In conclusion, 11 CC-PDX models were established. Based on the pathological and genomic findings, strong similarities were observed between the primary and PDX tumors. The gene expression in EVs was also similar between the two groups. Therefore, the PDX models could be potentially useful tools for translational cancer research. 

## Figures and Tables

**Figure 1 cancers-14-02969-f001:**
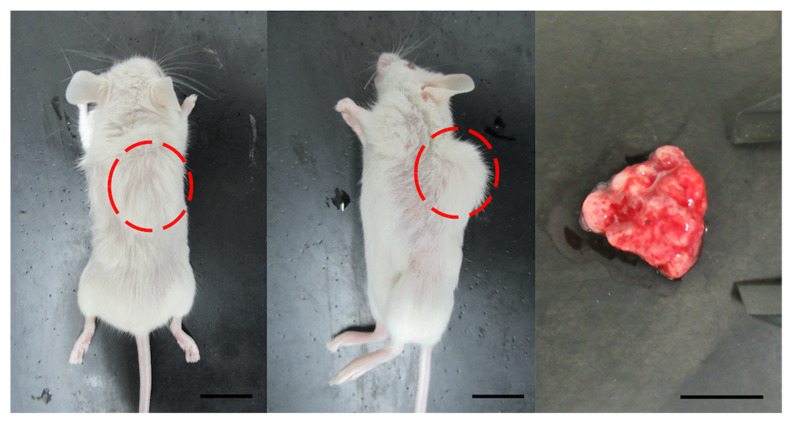
Development of tumors from a cervical cancer xenograft model (PDX71). **Left**, whole body image from the back. Solid tumor is evident (red circle). **Center**, whole-body image from lateral view. The tumor consists of a single nodule (red circle). **Right**, isolated tumor. Scale bar in all images = 10 mm.

**Figure 2 cancers-14-02969-f002:**
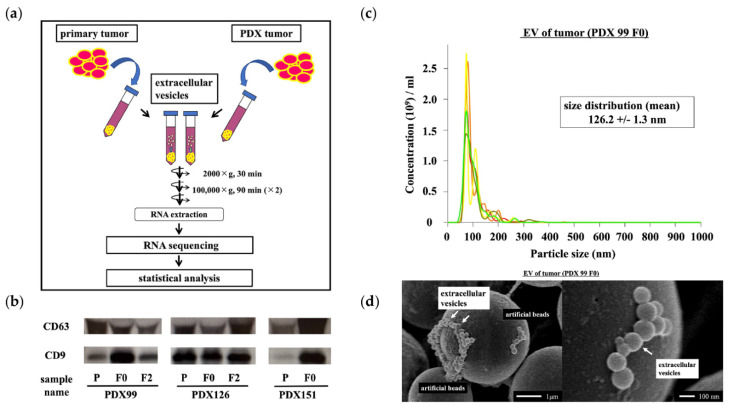
Isolation, characterization, and confirmation of extracellular vesicles (EVs). (**a**) EVs were isolated from the culture media of primary and patient-derived xenograft (PDX) tumors. Total RNA was extracted from the isolated EVs, and RNA sequencing was performed. The experimental results were statistically analyzed. (**b**) Western blot analyses were performed to detect exosomal marker proteins (CD9 and CD63) in vesicles released by primary and PDX tumors. Representative examples of bands from three independent experiments are shown. (**c**) The particle size distributions and concentrations of EVs were measured using NanoSight NS 300. (**d**) Representative images of EVs obtained using scanning electron microscopy.

**Figure 3 cancers-14-02969-f003:**
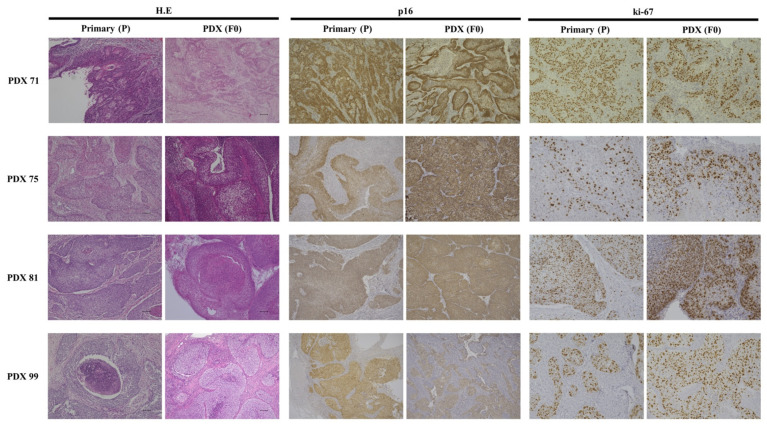
Pathological findings with immunohistochemistry of primary and PDX models (PDX 71, 75, 81, and 99). Tumor cells show angular and irregularly sized and shaped nests, anastomotic cord, and solid sheet interstitial infiltration. Nuclear pleomorphism and an increased mitotic count are observed in primary and PDX models. Immunohistochemical patterns are similar for p16 and ki67 between the primary and PDX tumors. Scale bar = 100 μm.

**Figure 4 cancers-14-02969-f004:**
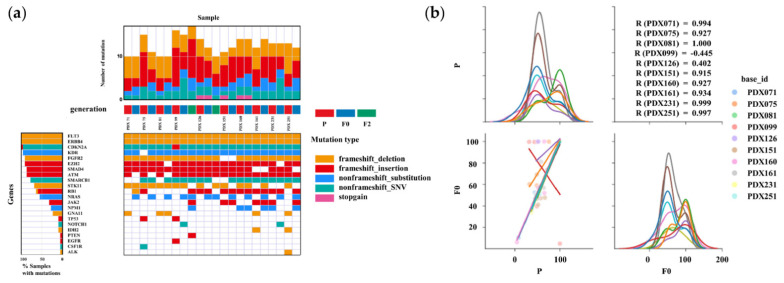
Summary of the relationships between somatic mutations in primary and PDX tumors based on DNA-seq results. (**a**) Heatmap analysis of the DNA profiling in tissues of the PDX and primary tumors using the AmpliSeq Cancer Hotspot Panel v 2. In all cases, gene mutations in primary (P) and PDX (F0 and F2) tumors were found to be highly correlated. (**b**) Variant allele frequencies (VAFs) of somatic mutations identified in P and F0 tumors. The lower-left graph shows a scatter plot and linear regression of the VAF levels in the P and F0 tumors. The diagonal graph shows the kernel density estimation (KDE). In most cases, P and F0 tumors show a high correlation whereas PDX99 tumor show a low correlation.

**Figure 5 cancers-14-02969-f005:**
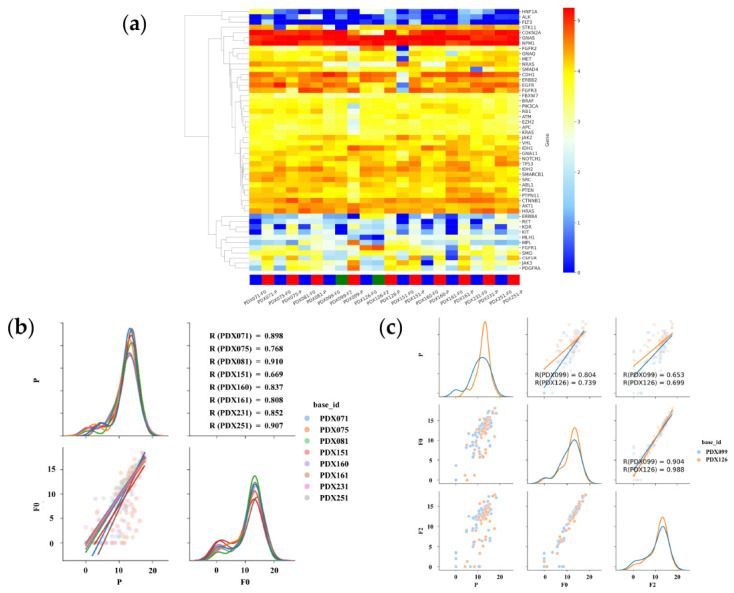
Summary of the relationship between gene expression in primary and PDX tumors based on RNA sequencing (RNA-seq). (**a**) The clustering and heatmap analysis of the mRNA profiling in tissues of the PDX and primary tumors using the AmpliSeq RNA Cancer Panel. In all cases, gene expression in primary (P) and PDX (F0 and F2) tumors are highly correlated. (**b**) Pair plot showing gene expression using the AmpliSeq RNA Cancer Panel in eight PDX (71, 75, 81,151, 160, 161, 231, and 251) models. The lower-left graph shows a scatter plot and linear regression of gene expressions in the P and F0 tumors. The diagonal graph shows the KDE. In all cases, gene expression in P and F0 tumors was found to be highly correlated. (**c**) Pair plot showing gene expression using the AmpliSeq RNA Cancer Panel in two PDX (99 and 126) models. The graph in the middle of the left column shows a scatter plot and linear regression of gene expression in the P and F0 tumors. The graph at the bottom of the center column shows a scatter plot and linear regression of gene expression in the F0 and F2 tumors. The lower-left graph shows a scatter plot and linear regression of gene expressions in the P and F2 tumors. The diagonal graph shows the KDE. In all cases, gene expression in P and F0 tumors was found to be highly correlated. Gene expression in F0 and F2 tumors is more highly correlated. Normalized data are converted to base 10 logarithms and z-scores.

**Figure 6 cancers-14-02969-f006:**
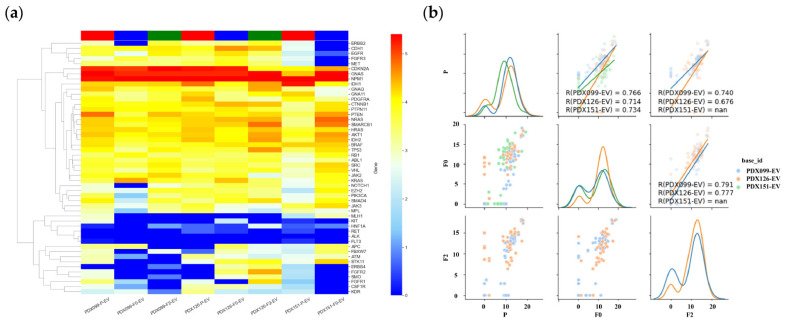
Summary of the relationship between gene expression in the EVs of primary (P-EV) and PDX (F0-EV, F2-EV) tumors based on RNA-seq. (**a**) Clustering and heatmap analysis of mRNA profiling results in P-EV, F0-EV, and F2-EV using the AmpliSeq RNA Cancer Panel. In all cases, gene expression in P-EV was highly correlated with that in F0-EV, F2-EV. (**b**) Pair plot showing gene expression using the AmpliSeq RNA Cancer Panel in 3 PDX (99, 126, and 151) models. The graph in the middle of the left column shows a scatter plot and linear regression of gene expression in the P-EV and F0-EV. The graph at the bottom of the center column shows a scatter plot and linear regression of gene expression in the F0-EV and F2-EV. The lower-left graph shows a scatter plot and linear regression of gene expressions in the P-EV and F2-EV. The diagonal graph shows the KDE. In all cases, the gene expression in P-EV and F0-EV is highly correlated. Gene expression in F0-EV and F2-EV tumors is more highly correlated. The normalized data were converted to base 10 logarithms and z-scores.

**Table 1 cancers-14-02969-t001:** Patient Characteristics of Tumorigenicity of Engrafted Tumors in Xenograft Models.

Number	PDX	Growth	Age (year)	SCC (ng/mL)	CA125 (U/mL)	Histological Type	FIGO	LVI	DSI	Tumor (mm)
1	6	No	70	1.8	21.5	adenocarcinoma	IB2	Yes	Yes	40
2	18	Yes	48	4.4	86.2	adenosquamous carcinoma	IB2	No	Yes	65
3	51	No	37	6.6	69.1	squamous cell carcinoma	IIIC1	No	No	17
4	55	No	46	2.4	7.4	squamous cell carcinoma	IIA1	No	No	35
5	71	Yes	42	1.9	9.6	squamous cell carcinoma	IB1	No	No	14
6	73	No	41	0.9	18.4	adenosquamous carcinoma	IIA1	Yes	Yes	23
7	75	Yes	44	125.7	28.0	squamous cell carcinoma	IIA2	No	Yes	54
8	81	Yes	33	26.9	8.9	squamous cell carcinoma	IIIC1	No	Yes	58
9	99	Yes	43	25.8	36.1	squamous cell carcinoma	IB2	No	Yes	60
10	126	Yes	62	5.2	15.0	squamous cell carcinoma	IIA1	No	Yes	32
11	143	No	38	26.6	30.5	squamous cell carcinoma	IIA2	Yes	Yes	40
12	151	Yes	49	8.8	50.6	squamous cell carcinoma	IB2	No	No	41
13	160	Yes	53	0.7	7.9	squamous cell carcinoma	IB1	No	No	14
14	161	Yes	68	8.3	17.5	squamous cell carcinoma	IIIC1	Yes	Yes	70
15	163	No	54	1.3	8.0	adenocarcinoma	IB1	No	No	32
16	215	No	49	1.3	11.7	adenosquamous carcinoma	IA1	No	No	5
17	229	No	33	1.8	23.3	adenocarcinoma	IB1	No	Yes	28
18	230	No	50	4.9	18.1	adenocarcinoma	IIA2	Yes	No	40
19	231	Yes	49	7.0	16.0	squamous cell carcinoma	IIB	Yes	Yes	40
20	251	Yes	38	2.6	14.6	squamous cell carcinoma	IIA2	No	Yes	60
21	269	No	53	10.3	11.4	squamous cell carcinoma	IIA1	Yes	No	33
22	270	No	56	13.0	15.5	squamous cell carcinoma	IB2	No	No	65

SCC, squamous carcinoma antigen; CA125, carbohydrate antigen 125; FIGO, The International Federation of Gynecology and Obstetrics; LVI, Lymphovascular invasion; DSI, Deep Stromal Invasion.

## Data Availability

The data that support the findings of this study are available on request from the corresponding author. The data are not publicly available due to privacy restrictions.

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
