# Peer review of "Validation of a Patient-Derived Xenograft Model for Cervical Cancer Based on Genomic and Phenotypic Characterization"

_cancers, 2022, doi:10.3390/cancers14122969_

Round 1

Reviewer 1 Report

Miyamoto, S. et al report that the establishment of 11 PDX models of cervical cancer with a 50% success ratio via subcutaneously injection of the tumor fragments from 22 patients into NOD.CB17-PrkdcSCID/J mice. Histological and immunohistochemical analyses revealed that there are no major Histological differences between the original and PDX tumors. Genetic analysis revealed that most of the gene mutations and expression related to carcinogenesis and infiltration were similar between the primary tumor and xenograft.

Overall, the manuscript is well written and the data are very solid even though the sample size is small due a physical limitation. Two major concerns need to be addressed to improve its value for clinical practices.

1. Even though most of the gene mutations and expression related to carcinogenesis and infiltration were similar between the primary tumor and xenograft, the differences need to be pointed out and discussed, which may involve in the pathway changes for downstream studies.

2. The final purpose for the establishment of PDX is to study the treatment response on mouse model for a better treatment regimen to a patient. Have the authors done any treatment to the PDX mice? At a minimum, a PARP1 inhibitor treatment should be done in PDX mice a mutation of BRCA1 or anti-EGFR treatment with a EGFR amplified. This result will demonstrated the real value of PDX model from bench to bed  

Author Response

We appreciate the time and effort provided by the editor and referees in reviewing our manuscript. We have addressed all issues indicated in the review report and hope that the revised version meets the journal's requirements for publication.

Response to Comments from Reviewer 1:

Comment 1: Even though most of the gene mutations and expression related to carcinogenesis and infiltration were similar between the primary tumor and xenograft, the differences need to be pointed out and discussed, which may involve in the pathway changes for downstream studies.

Response: We are grateful for this recommendation. Based on your suggestions, we have added the sentences on gene mutations to the Discussion. (page 12, line 476-482)

Comment 2: The final purpose for the establishment of PDX is to study the treatment response on mouse model for a better treatment regimen to a patient. Have the authors done any treatment to the PDX mice? At a minimum, a PARP1 inhibitor treatment should be done in PDX mice a mutation of BRCA1 or anti-EGFR treatment with a EGFR amplified. This result will demonstrated the real value of PDX model from bench to bed 

Response: PDX models must be useful for preclinical tests, as you have mentioned. Furthermore, several gene mutations that could be used as targets for therapy were identified in our study. However, we did not schedule drug evaluation tests in this study. Although the fragments obtained from PDX are reserved as frozen tissue for re-transplantation, the process is time- and cost-intensive. Unfortunately, we could not perform a drug efficacy test in this study. However, we intend to perform the same in our subsequent study. Based on your suggestions, we have added the sentences in the Discussion. We have also added the limitation. (page 12, line 483-512 page 13 line 516-519)

Reviewer 2 Report

Congratulations on your draft. I have only few things to add:

Minor revisions:

  1. Lacking simple summary
  2. Figure 2 and 3 should be merged together. There is no need to put so much emphasis on extracellular vesicles in this paper.
  3. The authors should test a drug on their model and report their data. As control they should use conventional methods.

Author Response

We appreciate the time and effort provided by the editor and referees in reviewing our manuscript. We have addressed all issues indicated in the review report and hope that the revised version meets the journal's requirements for publication.

Response to Comments from Reviewer 2:

Comment 1: Lacking simple summary

Response: We apologize for the absence of the simple summary. We have added the simple summary to the revised manuscript. (page 1, line 16-21)

Comment 2: Figure 2 and 3 should be merged together. There is no need to put so much emphasis on extracellular vesicles in this paper.

Response: Based on your suggestion, we have merged Figure 2 and Figure 3. (page 4, line 133-151; Figure 2)

Comment 3: The authors should test a drug on their model and report their data. As control they should use conventional methods.

Response: PDX models must be useful for preclinical tests, as you have mentioned. Furthermore, multiple gene mutations that could be used as targets for therapy were identified in our study. However, we did not schedule drug evaluation tests in this study. Although the fragments obtained from PDX are reserved as frozen tissue for re-transplantation, the process is time- and cost-intensive. Unfortunately, we could not perform a drug efficacy test in this study. However, we intend to perform the same in our subsequent study. Based on your suggestions, we have added the sentences in the Discussion. We have also added the limitation. (page 12, line 483-512; page 13 line 516-519)

Reviewer 3 Report

The authors engrafted patient tumor samples into NOD.CB17-PrkdcSCID/J mice and have shown that the primary patient samples and different generations of PDX samples have similar histopathological characteristics and mutational burden. In doing so, they have shed light on the better potential of PDX models as tools for translational research. The authors have designed their experiments well and employed appropriate techniques and executed their experiments very well. 

Minor comment.

1. For western blot data, include a loading control protein (beta-actin, tubulin or GAPDH),

2. Include the uncropped western blot data as supplemental data.

Author Response

We appreciate the time and effort provided by the editor and referees in reviewing our manuscript. We have addressed all issues indicated in the review report and hope that the revised version meets the journal's requirements for publication.

Response to Comments from Reviewer 3:

Comment 1: For western blot data, include a loading control protein (beta-actin, tubulin or GAPDH),

Response: Based on your suggestion, we have added a loading control protein ((beta-actin) for western blotting in the supplemental file. (supplemental figure 1)

Comment 2: Include the uncropped western blot data as supplemental data.

Response: Based on your suggestion, we have added the uncropped western blot data as supplemental data. (supplemental figure 1)

Round 2

Reviewer 1 Report

My concerns are nicely addressed. I recommend accepting it for publication.